METHODS

# Improved detection of microbiome-disease associations via population structure-aware generalized linear mixed effects models (microSLAM)

**Miriam Goldman** [1,2☉], **Chunyu Zhao**[2,3☉], **Katherine S. Pollard**[1,2,3*]

**1** Department of Epidemiology & Biostatistics, University of California San Francisco, San Francisco, California, United States of America, **2** Institute of Data Science & Biotechnology, Gladstone Institutes, San Francisco, California, United States of America, **3** Data Science, Chan Zuckerberg Biohub Network, San Francisco, California, United States of America

☉ These authors contributed equally to this work.
* katherine.pollard@gladstone.ucsf.edu

## Abstract

Microbiome association studies typically link host disease or other traits to summary statistics measured in metagenomics data, such as diversity or taxonomic composition. But identifying disease-associated species based on their relative abundance does not provide insight into why these microbes act as disease markers, and it overlooks cases where disease risk is related to specific strains with unique biological functions. To bridge this knowledge gap, we developed microSLAM, a mixed-effects model and an R package that performs association tests that connect host traits to the presence/absence of genes within each microbiome species, while accounting for strain genetic relatedness across hosts. Traits can be quantitative or binary (such as case/control). MicroSLAM is fit in three steps for each species. The first step estimates population structure across hosts. Step two calculates the association between population structure and the trait, enabling detection of species for which a subset of related strains confer risk. To identify specific genes whose presence/absence across diverse strains is associated with the trait, step three models the trait as a function of gene occurrence plus random effects estimated from step two. Applying microSLAM to 710 gut metagenomes from inflammatory bowel disease (IBD) samples, we discovered 56 species whose population structure correlates with IBD, meaning that different lineages are found in cases versus controls. After controlling for population structure, 20 species had genes significantly associated with IBD. Twenty-one of these genes were more common in IBD patients, while 32 genes were enriched in healthy controls, including a seven-gene operon in *Faecalibacterium prausnitzii* that is involved in utilization of fructoselysine from the gut environment. The vast majority of species detected by microSLAM were not significantly associated with IBD

**Data availability statement:** Reference genomes were obtained from the UHGG database version 2, available from the Mgnify FTP site (http://ftp.ebi.ac.uk/pub/databases/metagenomics/mgnify_genomes/) and also available from the European Nucleotide Archive under study accession ERP116715. Additional Faecalibacterium prausnitzii genomes were downloaded from NCBI. Supplemental Table 5 contains a full list of the Faecalibacterium prausnitzii NCBI accession numbers. Metagenomic sequencing data were obtained from NCBI. Supplemental Table 1 contains a full list of study accession numbers and sample accession numbers

**Funding:** This work was supported by the National Science Foundation (grant #1563159 to KSP), the National Institutes of Health (grant #HL160862 to KSP), the Chan Zuckerberg Biohub Network San Francisco (investigator support to KSP), and Gladstone Institutes (laboratory support and salary to KSP). KSP received salary from Gladstone Institutes, NSF (grant #1563159), and NHLBI (grant #HL160862). CZ received salary from Chan Zuckerberg Biohub Network. MG received salary from NHLBI (grant #HL160862). The funders had no role in study design, data collection and analysis, decision to publish, or preparation of the manuscript.

**Competing interests:** The authors have declared that no competing interests exist.

using standard relative abundance tests. These findings highlight the importance of accounting for within-species genetic variation in microbiome studies.

## Author summary

The species composition of the human gut microbiome differs significantly between individuals and is associated with various diseases. Many studies have sought to understand this relationship by examining the relative amount of each bacterial species within metagenomic sequencing data from sick and healthy individuals. However, this approach makes it challenging to pinpoint which genes and pathways of a disease-associated species might actually contribute to disease risk, and it misses species where only certain strains are disease associated. To overcome these challenges, we developed an R package, called microSLAM, that uses mixed-effects modeling to associate within-species genetic variation to host traits. In addition to testing if species abundance is associated with the trait, microSLAM performs two types of genetic association tests for each species: one for identifying strain-trait associations and another for identifying gene-trait associations. The gene tests account for the genetic relatedness of strains across hosts, making them particularly useful for detecting mobile genes. We applied microSLAM to hundreds of gut metagenomes from inflammatory bowel disease studies, identifying dozens of novel associations that were missed using relative abundance tests. MicroSLAM is a general modeling approach that can be applied to human traits beyond disease case/control studies and to microbiomes from other environments.

## Introducti Refer c15 attachment if you need cover image for this research on

The human body is home to a complex community of microorganisms, known as the microbiome, which encodes millions of genes [1]. The species composition of the microbiome differs significantly between individuals and is associated with host genetics, diet, immune system, and numerous human diseases [2–5]. As microbiome species evolve, individual lineages lose and gain genes through horizontal gene transfer [6,7] and other processes that create structural variation [8–10]. The resulting pangenome can be quantified from shotgun metagenomics data [11–13], which has revealed immense genetic diversity between and within human hosts [14]. Even when two people harbor the same microbial species, the cells within those populations are likely to perform different functions [10,15]. For example, prior studies identified many cases of variable virulence and antibiotic resistance [16,17], a set of pro-inflammatory genes from specific strains of *Ruminococcus gnavus* [18], a *Faecalibacterium prausnitzii* GalNAc utilization pathway

linked to with cardiometabolic health [19], and a strain of *Escherichia coli* with enhanced ability to live on the intestinal mucus that is associated with inflammatory bowel disease (IBD) [20]. These findings underscore the limitations of using species abundance alone to gain insight into host-microbiome interactions.

Many microbiome studies use pipelines such as HUMAnN [21] or MGnify [22] to map sequencing reads to known functions, including metabolic pathways, enzyme families, and gene ontology terms. Functional categories can be tested to determine if their presence or abundance is associated with a host trait. This approach effectively captures broad functional capabilities and gains power when functions are shared across species since mapped reads are pooled over species. However, poorly annotated or recently acquired genes—particularly mobile genetic elements and lineage-specific genes—may go undetected if they lack close homologs across species or are poorly represented in functional annotation databases. These invisible genes can play key roles in strain-level adaptations relevant to host health, such as antibiotic resistance, xenobiotic metabolism, or immune system interactions. Species-level gene-trait association tests can complement these function-based methods by revealing which organisms carry each trait-associated gene, ensuring that uncommon or specialized genes are not overlooked due to incomplete annotations or a narrow phylogenetic distribution. They provide a level of genomic resolution that aids downstream experimental validations and targeted interventions.

In this study, we consider two ways to leverage within-species pangenomic diversity to discover associations between the microbiome and a trait of the host, such as disease. The first is designed for when a species has a strain or group of related strains that predicts the trait. Identifying and isolating trait-associated strains facilitates experimental investigations into host-microbiome interactions, and strains enriched in healthy hosts have been proposed as components of probiotics and therapies [22–25]. Due to the systematic structure of bacterial genomes in which many genes have correlated presence/absence across strains–especially closely related strain–this approach will typically identify a large set of trait-associated genes. While any of these genes could be a good biomarker (e.g., for diagnosis or patient stratification), most of them are not good candidates for follow-up studies of causal mechanisms. Therefore, we also consider a second case in which one or a small number of individual genes predict the trait. Such associations are easiest to detect if the genes are rapidly gained and lost (e.g., mobile elements), so that they associate with the trait independently of evolutionary relationships amongst strains. Genes like this are still only biomarkers for the trait (i.e., statistical associations), but they are promising candidates for follow-up studies aimed at discovering causal mechanisms through which microbes modify host health and treatment responses.

To identify trait-associated microbiome strains and genes, we developed a statistical model that can be used to perform a metagenome-wide association study (MWAS) for any continuous or binary host trait. Building on the work done on generalized linear mixed-effects models from human genetics [26–30], this modeling approach uses gene presence/absence data from cohorts with metagenomic sequencing to first estimate a between-sample genetic relatedness matrix (GRM) for each microbiome species and associate this population structure with the host trait. Then, each gene in a species' pangenome is tested for its trait association after accounting for the relatedness of strains across hosts using random effects derived from the GRM. Our methodology is implemented in an open-source R package, called microSLAM, which can be used with quantitative and binary traits (including unbalanced case/control studies), scales to thousands of samples, and has a controlled type one error rate. The two tests in microSLAM enable researchers to detect new associations and to refine associations discovered using relative abundance.

To investigate the utility of microSLAM, we analyzed a compendium of 710 publicly available metagenomes from IBD case/control studies (S1 Table). IBD is an inflammatory condition of the gastrointestinal tract characterized by its persistence [31]. IBD afflicts roughly three million Americans [32], and its incidence has continued to increase in older adults in recent years [33]. The gut microbiome has long-standing links to IBD, including species abundance and gene associations [15,20,30,34–42]. Here, we combined MIDAS v3 pangenome profiling [11] with microSLAM to quantify associations between IBD and relative abundance [43–45], population structure, and gene presence/absence across 71 common

members of the human gut microbiome. These analyses identified 42 species with IBD-associated population structure and 56 significant gene families, which we interpreted at the pathway level within and across microbiome species. Tests based on relative abundance would have missed these associations.

## Results

### Overview of microSLAM

MicroSLAM is a statistical framework designed to perform MWAS using microbiome population structure-aware generalized linear mixed effects models. It was inspired by the application of mixed models in human genetics and was specifically developed to address the unique characteristics of microbiome data. By modeling both strain-level and gene-level effects, microSLAM ensures that microbial contributions to host traits are accurately identified while mitigating confounding effects of microbial population structure on gene-trait associations. Relative abundance testing can also be incorporated, allowing users to test for its association with the trait while adjusting for it in the strain-level and gene-level tests. The examples in this study focus on the human gut microbiome and IBD case/control cohorts, but microSLAM can be applied to any trait or environment. The microSLAM method is implemented as an open-source R package at: https://github.com/pollardlab/microSLAM.

### Adapting mixed modeling to microbiome genotypes

microSLAM is designed to analyze microbial gene presence/absence data (0/1), which differs from single nucleotide polymorphism (SNP) data from diploid organisms, typically represented as the number of copies of the derived allele (0/1/2). Gene presence/absence genotypes require a distinct approach to modeling genetic relatedness and performing association tests. To accomplish this, we extended the SAIGE mixed modeling approach [29] to (i) perform separate modeling and testing per microbiome species, (ii) compute species-specific genetic relatedness matrices (GRMs) using gene presence/absence with similarity metrics appropriate to binary data (e.g., pairwise Manhattan distance), and (iii) test each gene in each species' pangenome for evidence that gene presence associates with a host trait using the same score-based approach as SAIGE. MicroSLAM modeling is performed downstream of running a pangenome profiling tool, such as PanPhlAn 3 [21], Roary [12], or MIDAS v3 [11], to call gene presence/absence across a set of metagenomic samples. By treating gene presence as the genotype, microSLAM is distinct from other extensions of SAIGE that aggregate SNPs within regions or genes [46,47].

### microSLAM framework

microSLAM consists of three major components that are run for each microbial species and a host trait (Fig 1A, Methods): (1) construction of the GRM between samples; (2) estimation of variance components with a permutation test to determine whether the strain-level population structure is associated with the trait (τ test); (3) for each gene in the species' pangenome, estimation of the association between the trait gene presence/absence given the population structure and other covariates with a score test for statistical significance (β test).

Step 1: Constructing the GRM: A species-level n-times-n GRM $\Psi$ is computed using Manhattan-distance based on gene presence/absence data for n metagenomes.

Step 2: τ test (variance component estimation): The variance component τ represents the strain-level population structure that is associated with the host trait. microSLAM extends SAIGE's generalized linear mixed modeling approach to adapt to gene presence/absence data rather than SNP genotypes. For a binary trait the model in step 1 is:

$$logit(Y_i) = X_i\alpha + b_i + \epsilon_i, \ldots \quad b \sim N(0, \ \tau\Psi), \ldots \quad \epsilon \sim N(0, \ \sigma^2 I)$$

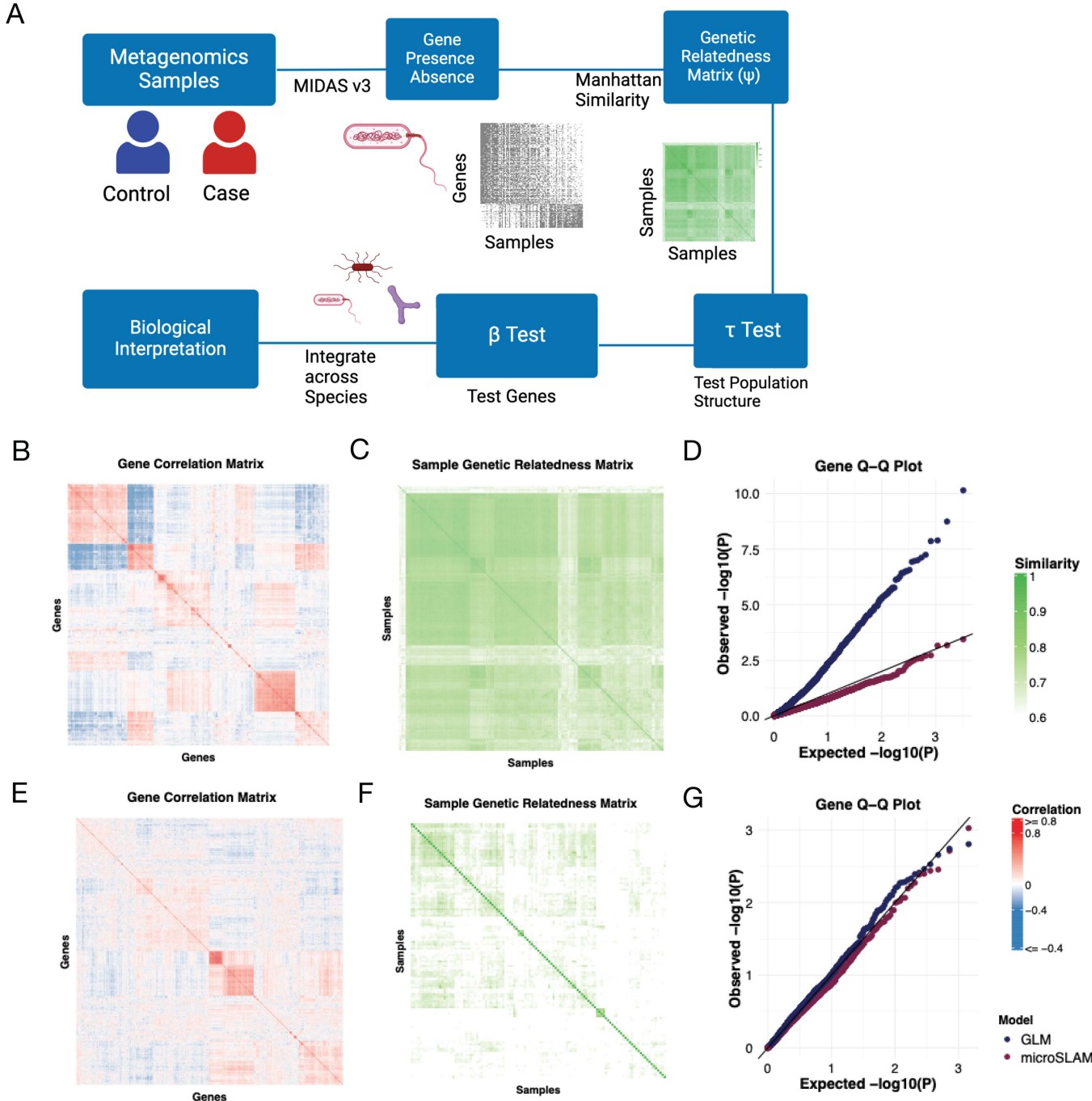

**Fig 1. MicroSLAM motivation and approach. A)** Flow chart of microSLAM modeling approach (diagram created in BioRender). **B – G)** Two bacterial species with different population structures. First row: High population structure species *Phocaeicola dorei* (260 IBD cases, 218 controls); Second row: Low population structure species *Blautia massiliensis* (73 IBD cases, 44 controls). **B&E)** Heatmap of gene-by-gene correlation matrix based on gene presence/absence across IBD samples, using the top 1000 most variable genes. Red: high positive correlation, Blue: high negative correlation. **C&F)** Heatmap of sample-by-sample GRM (1 minus Manhattan distance of gene presence/absence profiles). Dark green: high similarity, White: low similarity. **D&G)** Q-Q plot for p-values from tests of association between case/control status and presence/absence of individual genes in the pangenome. Tests are based on micoSLAM and standard logistic regression that does not adjust for population structure (glm). The diagonal line shows expected p-values under the null hypothesis of no association. Pangenome profiling for the metagenomes was done using MIDAS v3 [11].

where:

 $Y_i$: trait value for sample $i$ (e.g., disease/control)

 $X_i$: covariate matrix (e.g., age, sex, diet, species abundance)

 $\alpha$: fixed-effect coefficients for the covariates in $X_i$

 $b_i$: random effect for sample $i$ capturing strain-level population structure

 $\Psi$: species-specific GRM

 $\sigma^2$: residual variance

We then employ AI-REML (Average Information Restricted Maximum Likelihood) and PQL (Penalized Quasi-Likelihood) to estimate variance components and fit the generalized linear mixed model, following the same approach as SAIGE [29]. Instead of using a likelihood ratio test to assess the significance of population structure effects, microSLAM applies a non-parametric permutation test [48] to compute the p-value for $\tau$, providing a more robust evaluation of how significantly within-species population structure is associated with the trait.

A large $\tau$ value and small p-value indicate that cases are enriched for particular strains or lineages compared to controls. Identifying strain-trait associations is important because this improves the precision of research and therapeutics development based on cultured strains beyond simply picking a random strain from a trait-associated species, which may in fact not be one of the strains driving the species-level association. The genes jointly defining trait-associated strains (i.e., those positively or negatively associated with $b$) also provide signatures that can be used for predictive modeling and potential diagnostics.

Step 3: β test (gene-trait association test): The β-test evaluates how significantly a specific gene's presence/absence is associated with the trait while controlling for strain-level population structure and covariates. After fitting the model in the $\tau$ test once per species, it can be used as a null model to test if adding a gene's presence/absence significantly increases the likelihood. For a binary trait, the model for gene g is:

$$logit(Y_i) = X_i\alpha + \beta g_i + b_i + \epsilon_i$$

where variables are defined as above with the addition of:

 $g_i$: the presence/absence of the gene g in sample i

 $\beta$: effect size (log odds ratio for the trait given gene presence/absence)

The null hypothesis $H_0$: $\beta = 0$ (no gene-trait association) is tested using a score test. To mitigate inflated significance due to unbalanced case-control ratios, microSLAM employs saddlepoint approximation (SPA), following SAIGE's method to improve p-value accuracy [29].

For a quantitative trait, the logit link function in Step 2 and Step 3 is replaced with the identity link function to fit linear mixed effects models.

## Population structure in inflammatory bowel disease gut microbiomes

We compiled 710 publicly available gut metagenomes from five inflammatory bowel disease (IBD) case/control studies (S1 Table) and performed pangenome profiling of them using MIDAS v3. There were 71 species with sufficient sequencing coverage to analyze within species genetic variation. After dropping gene families that are nearly always present or nearly always absent (Methods), we had an average of 2,254 gene families per species and a total of ~160 thousand across species. In some species, such as *Phocaeicola dorei (P. dorei),* many gene families are co-evolving and show a high correlation in their presence/absence across hosts (Fig 1B). In turn, we see two distinct subgroups of strains in the GRM (Fig 1C). This high level of structure might be the result of selection pressures, drift, or a recent population expansion. When we perform MWAS for all *P. dorei* gene families using logistic regression (glm), we observe that most genes are significantly associated with IBD case/control status (Fig 1D). This inflation is similar

 

to the well-known problem in human genetics in which ancestry-associated variants are all highly significant when genetic ancestry differs between cases and controls [49]. In contrast, the gene-level test in microSLAM does not show inflation, because our model adjusts for population structure when testing individual gene families for disease associations. We therefore hypothesize that inflation is a consequence of high population structure resulting from a high correlation between gene families. Supporting this, *Blautia massilensis* does not have many genes that are correlated (Fig 1E) and shows less structure in its GRM (Fig 1F). Accordingly, the glm p-values do not show inflation, and the microSLAM output is very similar to that of glm.

These results suggest that if we wish to identify individual gene families with unexpectedly high associations with a host trait given the species' GRM, the mixed modeling approach in microSLAM provides a way to adjust for population structure across hosts, just as mixed models have enabled human geneticists to account for confounding from genetic ancestry. However, population structure is not necessarily a confounder in microbiome research, and it may also be of interest to identify trait-associated strains, defined by the presence and absence of many gene families relative to other strains. These genes would not be significant in the microSLAM β test, because they are highly correlated with population structure. For this reason, microSLAM also includes a strain-level test, the τ test.

Consider, for example, *Ruminococcus B gnavus (R. gnavus),* a species that has long been associated with IBD [18,50,51]. The *R. gnavus* GRM shows two distinct groups when hosts are sorted based on their $b_i$ values (Fig 2A). One of these groups only contains IBD individuals, while the other is split between IBD and controls. The $b_i$ estimates better separate cases and controls than do the first two principal coordinates of the gene presence/absence matrix (Fig 2B). Not surprisingly, when we apply the microSLAM τ test to *R. gnavus*, we obtain a large and statistically significant measure of association (τ = 3.92, permutation p-value = 0.0001; Fig 2C), and the resulting model can classify IBD cases with high accuracy (ROC AUC = 0.969; Fig 2D). Now, if we look at the genes that are most highly correlated to the estimated $b_i$ values of the samples (correlation to $b_i \geq 0.5$), we identify 186 out of the ~1200 non-core, non-rare genes used in the analysis that are all nearly equally associated with IBD (Fig 2E). None of these genes are significantly associated with IBD after adjusting for population structure. Thus, we were able to identify several hundreds highly correlated genes that form a predictive signature for the *R. gnavus* strains present in IBD patients versus controls. These observations illustrate the importance of including the τ test in microSLAM.

## MicroSLAM controls false positive rates and increases specificity in simulations

The examples in Figs 1 and 2 suggest that microSLAM's τ test can detect strain-trait associations when species have a high degree of population structure and its β test may control false positive gene-disease associations better than a standard glm, albeit somewhat conservatively. But the ground truth is unknown in real data. Hence, we designed a series of simulations to assess the performance of both of microSLAM's tests. Our simulation strategy leveraged the IBD compendium in order to capture the range of patterns observed in real data, while varying parameters such as effect size and sample size. For the β test, we compared microSLAM to glm in order to evaluate the effects of adjusting for population structure via the random effects $b_i$.

First, we evaluated the τ test. To the best of our knowledge, there is no other method that performs this type of association test, so we did not compare microSLAM to alternative approaches. To quantify type 1 error (i.e., false positive rate), we simulated gene presence/absence matrices with core and accessory genes, as well as a set of strain-specific genes, and for each host a binary trait was simulated independently of the gene presence/absence matrix so that the GRM is not associated with the trait (τ test simulation simulation 1; Methods). We investigated a sample size of 100 hosts, which is on the low end of what we observed for species in the IBD compendium, and repeated the simulation 1000 times, keeping track of how many iterations had a permutation p-value < 0.05. We observed a false positive rate of 0.054, which is very close to the expected value of 0.05. This indicates that the false positive rate of microSLAM's τ test is approximately correct.

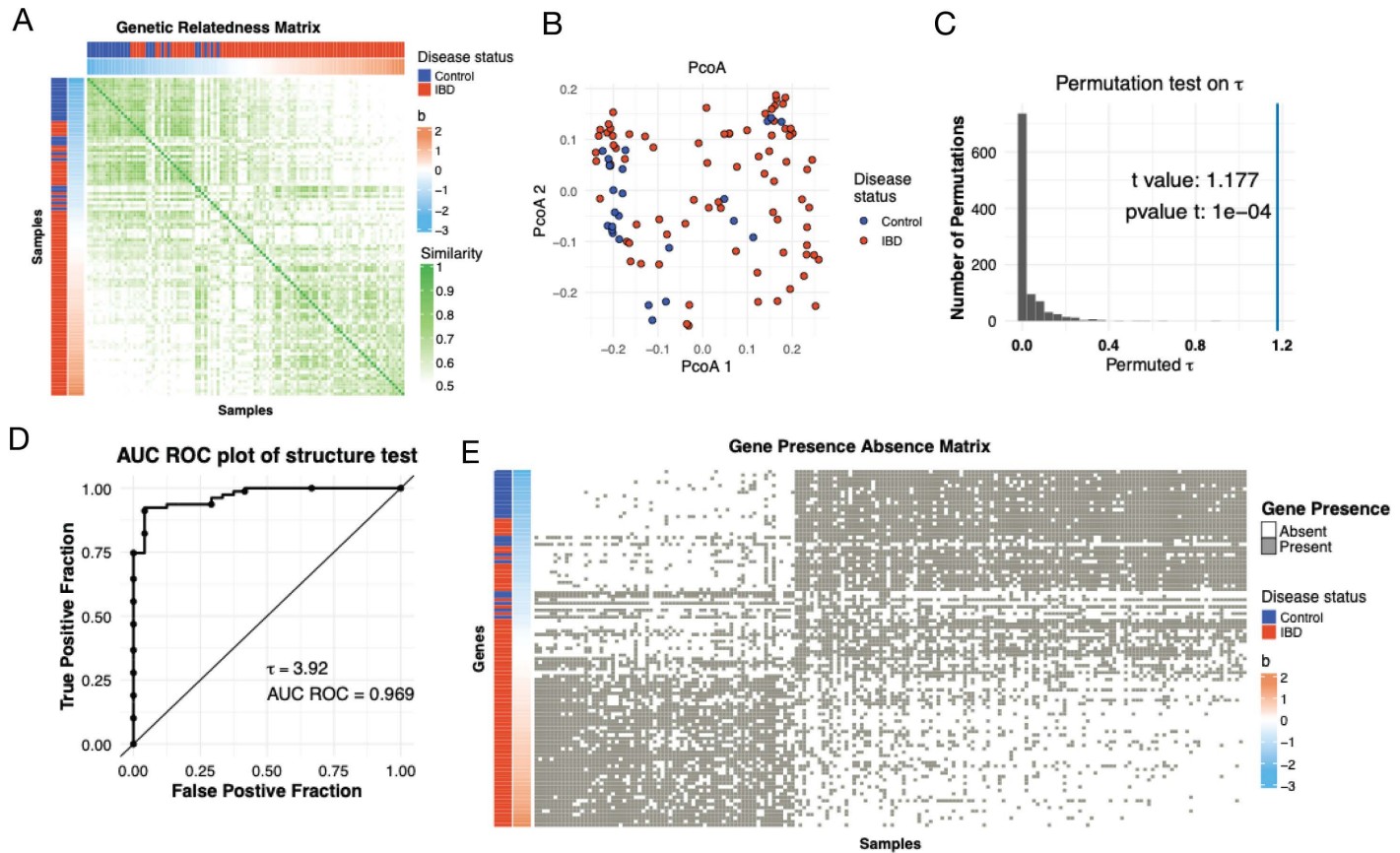

**Fig 2. MicroSLAM detects both strain and gene associations. A)** The GRM for *Ruminococcus B gnavus* with hosts sorted by their estimated *b* values and annotated by their disease status. **B)** PCoA from the *R. gnavus* gene presence/absence colored by host disease status (as in A). **C)** Histogram of permutation test statistics (*t*-values) from the τ test for *R. gnavus.* The line denotes the observed value of *t.* **D)** ROC plot for the microSLAM τ test model for *R. gnavus.* The statistic τ quantifies population structure. **E)** Gene presence/absence plot for a subset of genes associated with the random effect *b* for *R. gnavus.* Samples are ordered by *b* and annotated by their disease status.

To evaluate the power of the τ test, we modified the prior simulation so that the trait depends on presence/absence of a particular strain (τ test simulation 2; Methods). We varied the strength of the strain-trait association (odds ratio) and explored sample sizes ranging from 60 to 250. As expected, power increases with the odds ratio and sample size (Fig 3B). MicroSLAM achieves ~80% power at an odds ratio of 1.5 with 250 samples, whereas an odds ratio greater than 2.0 is needed for similar power with only 60 samples. These results provide practical guidelines for the expected performance of the τ test.

Next, we investigated the type 1 error and power of microSLAM's β test compared to glm, which is the standard approach used in the literature. We considered the case where gene presence/absence is not associated with a binary trait, but it is associated with population structure, and hence genes are correlated with each other. To do so, we simulated gene presence/absence using principal components of the observed GRM for each of the 71 species in the IBD compendium (β test simulation simulation 1; Methods). MicroSLAM controlled the false positive rate below 0.05 for all but two species where it is exactly 0.05 (*Dorea A longicatena, Roseburia sp900552665*). In contrast, the glm without a random effect adjusting for population structure failed to do so for all but 9 species (*Faecalibacterium prausnitzii, Bifidobacterium adolescentis, Bariatricus comes, Blautia A faecis, Faecalibacillus intestinalis, Gemmiger qucibialis, Akkermansia muciniphila,*

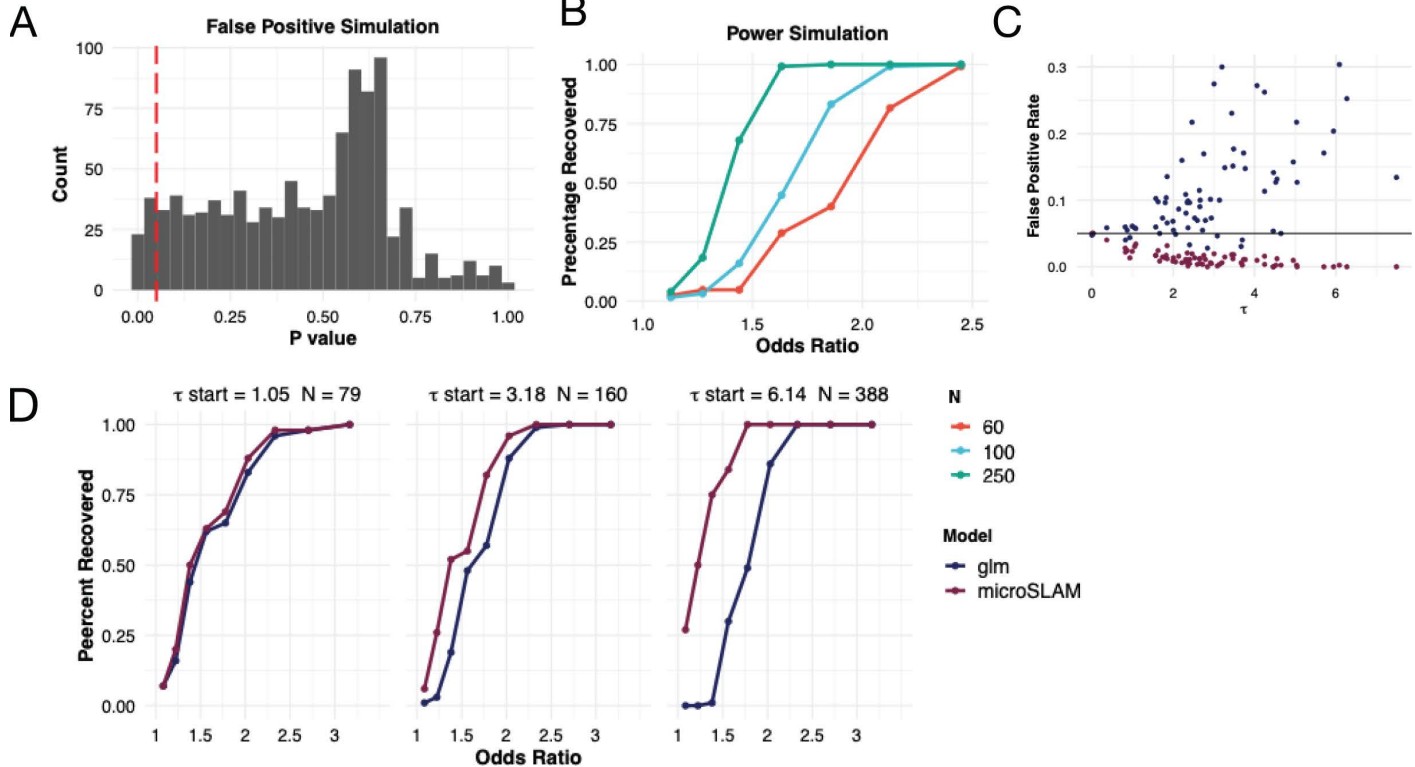

**Fig 3. Simulations show that microSLAM improves power and false positive rates. A)** The false positive rates of the τ test of microSLAM were estimated using simulations with varying GRMs but no trait associations. We simulated gene presence/absence and GRMs for the 1000 iterations (τ test simulation 1; Methods). A histogram of p-values for the τ tests shows that the percentage of tests with a p-value < 0.05 is 5.4%. **B)** Power of the τ test for simulations with a range of values for the odds ratio of the simulated y compared to presence of the trait-associated strain (τ test simulation 2; Methods), repeated for different numbers of samples (N). **C)** False positive rates of the β tests for glm and microSLAM were estimated using simulations with varying levels of population structure (τ) but no trait associations. We simulated gene presence/absence using the GRMs for the 71 species in the IBD compendium (β test simulation 1; Methods). The false positive rate increases with $\tau$ for the glm and is generally above the targeted level (0.05; horizontal line), while it decreases and is generally below 0.05 for microSLAM. **D)** Power for 3 simulated species with different τ values and numbers of samples (N). For a subset of genes, presence/absence is simulated based on the trait using a range of odds ratios; other genes have presence probabilities that do not depend on the trait (β test simulation 2; Methods).

*Roseburia sp900552665, Acetatifactor sp900066565*), a failure rate of 87.3% (62/71). In addition, as the estimated τ increased, the false positive rate of the glm dramatically increased while the false positive rate for microSLAM decreased slightly (Fig 3C).

To explore if this conservative control of the false positive rate affects the power of microSLAM's β test, we performed simulations where 100 true positive genes are added to the previously stimulated genes, meaning that they have a presence/absence pattern that is associated with the simulated trait (β test simulation 2; Methods). We varied the strength of the association (odds ratio) and evaluated power at an empirical false positive rate of 0.05 (calculated using the non-trait-associated genes). These analyses show that microSLAM consistently has either the same or higher power than the glm at the same false positive rate (Fig 3D), with the difference between methods being most pronounced with a higher number of samples and a high degree of population structure (τ) (S1 Fig). In order to understand exactly what types of genes lead to the higher false positive rate we simulated a data set completely *de novo* without using the IBD compendium (β test simulation 3; Supplement). This showed that the strain-associated genes tended to lead to an increase in false positives for the glm, while microSLAM was able to differentiate the true positives from genes linked to

the strain but not directly associated with the trait (S2 Fig). Thus, microSLAM's β test has better specificity than does a standard glm.

**MicroSLAM reveals IBD associations across 71 gut microbiome species**

We next sought to examine associations in our IBD compendium using microSLAM's population structure and gene tests. First, we performed the standard species-level analysis in which the relative abundance of each gut species (quantified using Kraken2 and Bracken [45–47]; Methods) is tested for association with IBD case/control status using logistic regression, adjusting for host age, which was consistently reported across studies. Medications and other clinical covariates are also important confounders but unfortunately were not provided in the publicly available datasets. For comparability, we used the same 71 species and species-specific number of samples that had sufficient coverage to perform microSLAM within-species association tests. We found that 3/71 species (4%) had significant relative abundance associations (localFDR < 10%; Fig 4A): *Lachnospira rogosae_A, Flavonifractor plautii,* and *Alistipes putridinis.* If the full set of 100 species and samples are utilized, the proportion with relative abundance associations rises to 18%.

Focusing on the 71 species subset, we next ran microSLAM using each species' gene presence/absence matrix and corresponding GRM (quantified using MIDAS v3 [11]; Methods). Because we observed significant variations in species abundance across studies, we used the MIDAS v3 genes flow to compute the median coverage of 15 universal, single-copy genes ("marker depth") in each sample (S3 Fig). This statistic is correlated with species relative abundance (by Kraken2 + Bracken), while also capturing cohort-to-cohort variation beyond abundance (e.g., sequencing depth, unmeasured confounders). Therefore, we added marker depth as a covariate in microSLAM as a way to indirectly account for cohort-specific batch effects without explicitly controlling for study name, thereby retaining statistical power (one parameter versus four parameters) while still addressing inter-study variation. Age was also included as a covariate. After including these covariates, the random effects ($b_i$) were not strongly associated with study (S3 Fig).

We used microSLAM's τ test to identify species whose population structure is associated with IBD case/control status. At localFDR < 10%, 56/71 species (69%) were significant (Fig 4A), meaning that cases and controls tend to harbor distinct strains consistently across studies. These included two out of the three species with relative abundance associations, indicating that specific strains of these two species may be responsible for differential abundance between cases and controls and leaving only one species (*Alistipes putridinis*) with a significant species-wide IBD-abundance association.

In addition to assessing the statistical significance of τ via a permutation test, we also report the area under the curve (AUC) for the receiver operating characteristic (ROC) from the τ test model. This shows how well the population structure component is able to separate the cases from the controls. The AUC is calculated within the same training data, because the random effects *b*, which are per-host parameters (i.e., on per subject) generated from the GRM, are unknown for new hosts and hence the fitted model does not generalize beyond the training set. Overall, the AUC from the τ tests was quite high; 55 species had AUC over 0.9. Class *Clostridia* tended to have the highest AUC values and a smaller variance in AUC values compared to *Bacteroidia* (Fig 4B). In addition to *R. gnavus* (Fig 2), species with significant τ tests included *Agathobacter rectalis* (previously found to be related to IBD under certain conditions [48]) and *Phocaeicola coprocola* (formally *Bacteroides coprocola*, which has been shown to have a relationship with ulcerative colitis [52]). In both of these species, there were no significant genes with the β test, but with the information from the τ test genes differentiating IBD-associated strains can be identified.

To investigate specific gene families associated with IBD case/control status, above and beyond the genes that define IBD-associated strains, we next applied microSLAM's β test. Across the 71 species, 53 genes from 20 species showed significant associations after adjusting for population structure (localFDR < 20%, which is the threshold with optimal lift and somewhat more lenient than the 10% threshold used for the other two tests). Of these species with genes that had significant β tests, three did not have significant relative abundance or population structure associations (Fig 4A). Conversely,

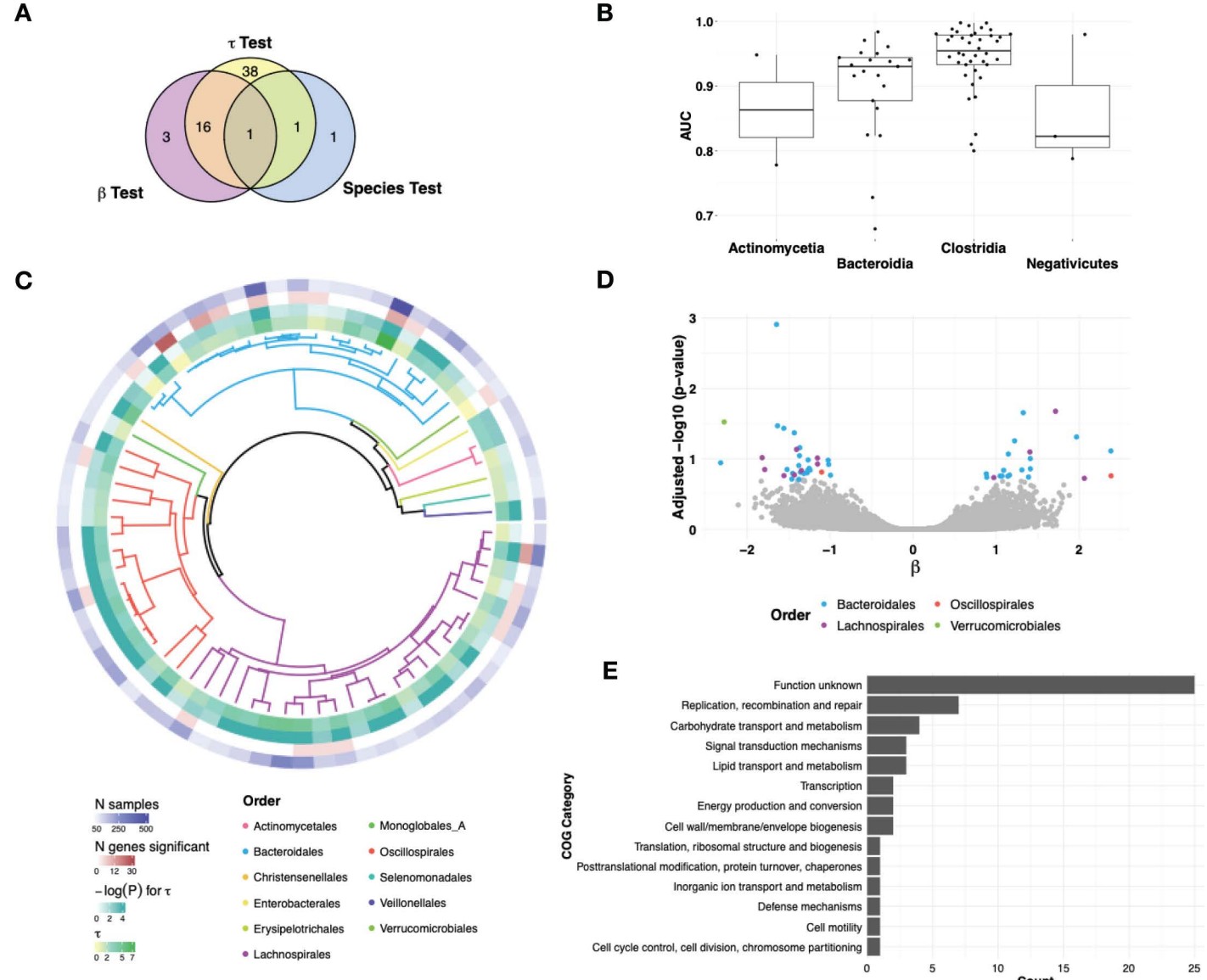

**Fig 4. MicroSLAM identifies novel IBD associations.** We analyzed all 71 species in our IBD compendium for three types of associations with case/control status: relative abundance (Kraken2 + Bracken: amount of the species predicts disease), population structure (microSLAM τ: strain predicts disease), and gene family (microSLAM β: gene presence/absence predicts disease). **A)** Venn diagram showing the number of species with significant IBD associations of each type. For genes, we counted the species if it had at least one significant gene family; species varied in the number of hits (S2 Table). All tests are localFDR adjusted for multiple testing. **B)** Boxplots showing the AUC ROC from τ test models for all 71 species, stratified by bacterial class. **C)** UHGG species tree for all 71 species, colored by order. The τ value, p-value for τ test, number of significant genes, and number of samples for each species are plotted in the outer rings. **D)** Volcano plot for β tests with significant genes (localFDR < 0.2) colored by bacterial order. (S3 Table) **E)** Bar plot of COG categories for the 83 genes with significant β tests.

38 species only had significant τ tests, and of those 25/38 (66%) are from class *Clostridia*. These results underscore the unique information captured by each of microSLAM's tests.

Having analyzed IBD associations at the species, strain and gene level, we integrated these results across the 71 species to look for phylogenetic trends (Fig 4C). Out of the 71 analyzed gut species, 11 (spread across the phyla *Firmicutes*

*A*, *Firmicutes C*, *Bacteriodota*, *Actinobacteria, and Verrucomicrobiota*) had no significant IBD associations, possibly due to a lower number of samples (N < 100 for ten species). As noted above, of the three species with relative abundance associations, two were detected with one or both of microSLAM's tests (Fig 4A), suggesting that relative abundance differences are often accompanied by differences in gene content. Looking across the phylogenetic tree, Lactobacillus species tend to have the least IBD-associated population structure (low values of τ), although there is a subclade of two species with higher τ values. On the other hand, Oscillospirales tend to have high values of, and most species in this order do not have any significant genes. Finally, Bacteroidales stands out as the order with the most significant genes (32/53), consistent with species in this order having many mobile and accessory genes [53].

To further explore the functions of genes identified by microSLAM's β test (Fig 4D), we looked at the functional annotation provided by MIDASDB [11]. As expected, many (47%) of the 53 significant genes had no functional annotation. For example, 25/53 are in the EggNOG COG category "Function Unknown". The remaining annotated genes were too few in number to perform well-powered enrichment analyses, but we did note several interesting trends (Fig 4E). The most common COG, encompassing 7 genes from 4 species, was "Replication, recombination and repair". Four genes were annotated as carbohydrate transport and metabolicm, 8 genes were annotated as putative plasmids by geNomad [54], among which 4 "unknown function" by EggNOG. Considering these annotations and the fact that these genes remain significant beyond the overall population structure of their species, we conclude that many of the significant genes identified by the β test are likely associated with mobile genetic elements.

## Seven-Gene GRF operon is a structural variant in *Faecalibacterium prausnitzii*

One significant gene identified by microSLAM was annotated as "subunit D of the fructoselysine/glucoselysine phosphotransferase (PTS) system" by BlastKOALA [55] (S4 Table). It was negatively associated with IBD case status in *Faecalibacterium prausnitzii D* (UHGG species id 102272), a species whose relative abundance is positively associated with IBD in our compendium. This hit intrigued us, because *F. prausnitzii* is a well-studied bacteria with roles in short-chain fatty acid metabolism and inflammation [56,57]. Predicting a new molecular mechanism underlying this host-microbe interaction would enable future functional studies (e.g., in gnotobiotic mice) and potentially could be useful for developing diagnostics, dietary interventions, or other therapies.

To explore this gene family, we first compiled 85 high-quality and diverse *F. prausnitzii* genome assemblies from NCBI (S5 Table) and clustered them into eight clades (Fig 5A; Methods). We observed that seven genes (plus occasionally an eighth gene) were consistently found together, with a conserved order and orientation across 53% of the NCBI *F. prausnitzii* genomes (49/85) (S4 Fig). Annotations suggest that these genes encode a fructoselysine/glucoselysine PTS system operon. Having established that this operon is variably present across distantly related *F. prausnitzii* strains, we expanded our search to include all high-quality *F. prausnitzii* genomes available in the United Human Gut Microbiome Database (UHGG v2) [1]. This analysis indicated that the complete seven-gene operon is present in all nine *F. prausnitzii* clades, with between ~3% and ~24% of genomes per clade containing the operon (Table 1).

Since genes can be syntenic without being functionally related, we conducted further analysis to determine the relationship between the seven-gene operon in *F. prausnitzii* and the well-characterized GFR operon in *Salmonella Typhimurium 14028s* [58]. We successfully mapped five genes from the *F. prausnitzii* operon to the corresponding *S. Typhimurium* operon (*gfrABCDF*) (Fig 5B). Notably, the *gfrE* gene, encoding a deglycase that cleaves glucoselysine 6-phosphate, is absent in *F. prausnitzii*. In addition, the operon in *F. prausnitzii* includes a gene without homology to any gene in the GFR *S. Typhimurium* operon. The regulatory genes, which are located at the start of the operons, also differ between the two species. We hypothesize the seven-gene *F. prausnitzii* operon identified in our analysis functions solely as a fructoselysine PTS system (Fig 5C).

Fructoselysine is a spontaneous product of Amadori rearrangements, and its presence in the human gut environment can promote the growth of bacteria capable of importing and using this carbohydrate as an energy source [59,60]. In *F.*

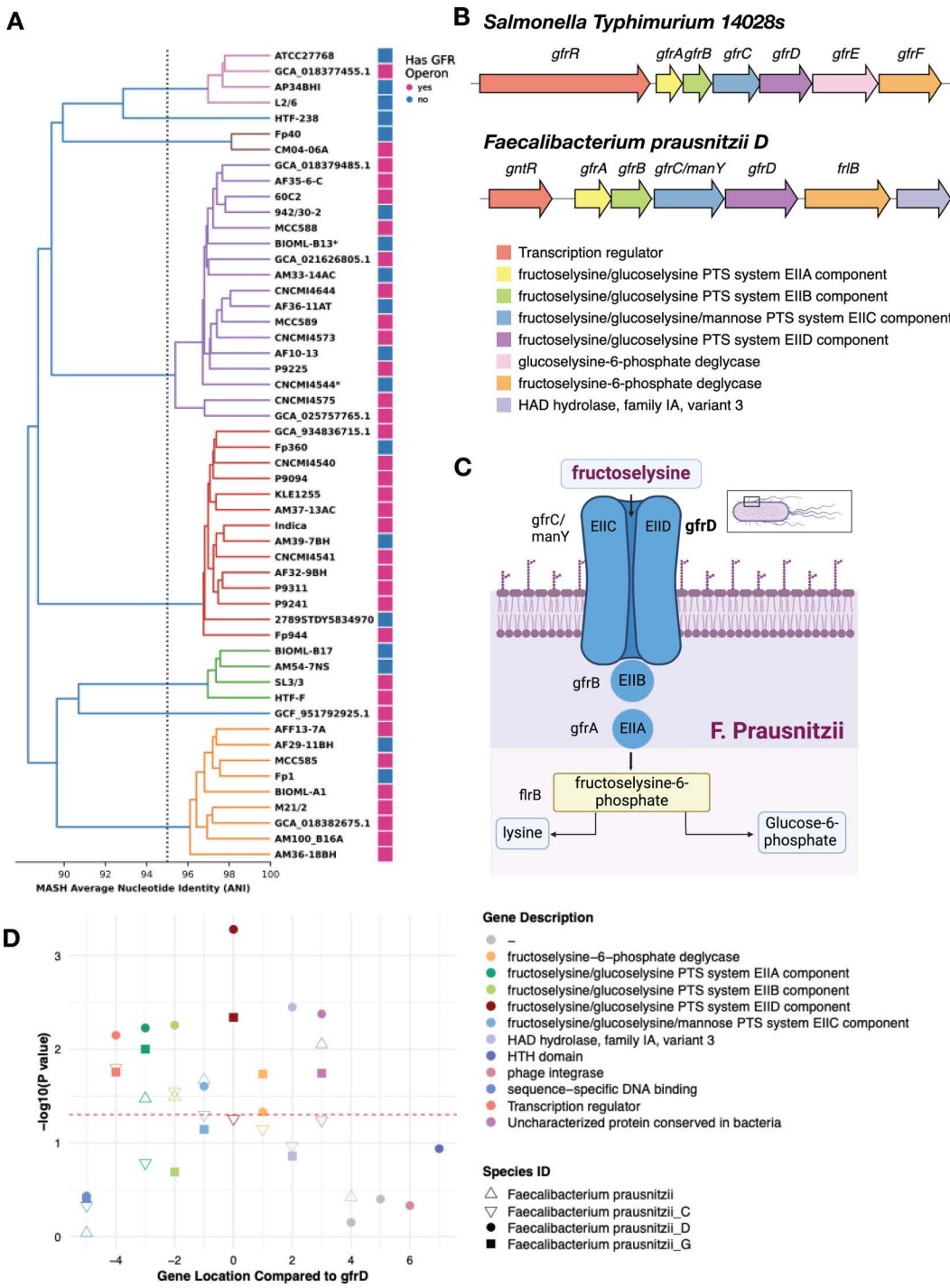

**Fig 5. Investigation of *F. prausnitzii* fructoselysine PTS system operon. A)** 52 representative genomes selected from NCBI and colored by the dRep secondary cluster (Selection of *Faecalibacterium prausnitzii* genomes, Methods). **B)** Comparison of *S. Typhimurium* operon to operon in *F. prausnitzii D*. **C)** Graphic of the *F. prausnitzii* fructoselysine PTS system operon and its products (made in BioRender). **D)** P-values for *F. prausnitzii* fructoselysine PTS system operon genes in microSLAM β tests across the four *F. prausnitzii* species defined by UHGG. The flanking genes are much less significant than the genes within the operon. Subunit D (most significant gene in microSLAM analysis) is located at 0, and all other indices are relative to this gene.

**Table 1. A seven-gene operon present in all nine *F. prausnitzii* clades in UHGG v2.**

| Species ID | Species name | Annotated as plasmid | Not annotated as plasmid | Total genome counts | Proportion with operon present |
| --- | --- | --- | --- | --- | --- |
| 100022 | F. prausnitzii C | 30 | 91 | 1258 | 0.096 |
| 100039 | F. prausnitzii H | 1 | 5 | 221 | 0.027 |
| 100195 | F. prausnitzii E | 6 | 4 | 244 | 0.041 |
| 101255 | F. prausnitzii F | 2 | 4 | 55 | 0.109 |
| 101300 | F. prausnitzii | 119 | 223 | 1446 | 0.237 |
| 102272 | F. prausnitzii D | 88 | 177 | 1280 | 0.207 |
| 102274 | F. prausnitzii I | 5 | 0 | 179 | 0.028 |
| 102545 | F. prausnitzii G | 124 | 127 | 2891 | 0.087 |
| 102619 | F. prausnitzii J | 25 | 32 | 236 | 0.242 |

*prausnitzii*, this is performed by a seven-gene operon encoding proteins that phosphorylate the substrate while transporting it across the bacterial cell membrane, making it available as a source of carbon. While only subunit D of this operon was significant in microSLAM's gene test after accounting for multiple comparisons, all of the genes in the operon had unadjusted p-values less than 0.05 and flanking genes in the *F. prausnitzii* reference genome did not (Fig 5D). Variability in gene detection from shotgun metagenomics data is a likely source of the difference in significance for subunit D versus the other genes. MicroSLAM analysis for three other *F. prausnitzii* species in the IBD compendium did not yield any significant genes, primarily due to inadequate sample sizes, which restricts the statistical power of microSLAM's β test. Nonetheless, the gene presence/absence matrices for these species are consistent with the operon being variably present and depleted in IBD cases.

Altogether, these results suggest that the genes in this *F. prausnitzii* PTS operon are co-evolving in terms of their presence/absence across hosts, potentially independently of neighboring genes, and that the presence of this operon is more common in healthy hosts. Our data also suggest that the fructoselysine PTS system operon could be a mobile genetic element. Supporting this possibility, many NCBI and UHGG contigs carrying this operon are predicted by geNomad [54] to be plasmids (Table 1). We also observe sequences associated with mobile elements and horizontal gene transfer (HGT) in the genomic context surrounding the operon. These are computational predictions only, and no plasmids have been previously reported in *F. prausnitzii*. We therefore checked for the operon in the other 70 species in our microSLAM analysis, detecting it in strains from two other phyla: *Gemmiger qucibialis* (species_id 103937) and *Faecalibacterium sp90053945* (species_id 103899). It is also known to be present in *Escherichia coli*, *Bacillus subtilis* and *Agrobacterium tumefaciens Ti plasmid [60]*. While not conclusive, these data are consistent with HGT. Regardless of the mechanism of acquisition or loss, the variable presence/absence of the operon across *F. prausnitzii* strains indicates that certain lineages of this species can acquire and utilize fructoselysine, thereby enhancing their adaptability and competitiveness in the dynamic gut ecosystem relative to strains without the operon.

## Discussion

In this paper we introduce microSLAM, a method that implements population structure-aware MWAS. We focused on case/control study designs, but microSLAM can also be applied to quantitative traits. Going beyond standard species relative abundance tests, microSLAM enables identification of specific strains and gene functions driving species-trait associations, as well as detection of novel trait associations not detectable at the species level. Using a generalized linear mixed modeling approach, we are able to include information about the genetic relatedness of the microbiomes across samples (GRM) and to model the association of this population structure with host traits (τ test), while adjusting for other covariates. In addition to strain-trait associations, microSLAM identifies gene-trait associations (β test). To be detected after adjusting for population structure, trait-associated genes must be evolving somewhat independently of the genes

that distinguish strain lineages, with mobile genetic elements being particularly strong signals in the cohorts we analyzed. Trait-associated genes are high-confidence candidates for studying causal mechanisms.

Through realistic simulation studies, we demonstrated that microSLAM controls Type 1 error and has reasonable power in cohorts with more than one hundred samples. Compared to standard glm, microSLAM's gene-level β test controls false positives much more effectively, especially in species with notable population structure. When there is a significant population structure as well as a subset of genes that are more related to the phenotype than the strain signal, we showed that microSLAM increases specificity compared to glm. By providing microSLAM as an open-source R package, we provide a new tool for researchers to probe microbiome-host interactions with strain- and gene-level resolution. We focused on applying microSLAM to the human gut microbiome to identify associations for IBD, but our R package is directly applicable to additional host traits and to additional environments where metagenomics data enables the computation of gene presence/absence matrices.

In this study, we put together a metagenomic compendium of IBD samples. Analyzing this data with microSLAM, we discover a wide variety of population structures within human gut metagenomes. We identified 56 species with a population structure related to IBD. In addition, after adjusting for population structure, 32 microbial genes are significantly enriched in healthy subjects, and 21 are enriched in IBD patients. From the genes enriched in healthy subjects, we identified a *F. prausnitzii* fructoselysine PTS system operon that is present in all clades of this species, but in only a minority of genomes within each clade, suggestive of being a mobile genetic element or other rapidly lost/gained structural variant. The presence/absence of this operon may confer distinct metabolic advantages to different strains, including the ability of carriers to utilize fructoselysine as an energy source [59]. The potential impact on human health could be significant, given that *F. prausnitzii* is one of the most prominent butyrate producers in the human gut [34,61,62]. This might also lead to a greater resilience of the gut microbiota, offering enhanced protection against pathogenic bacteria and reducing risk of chronic disease. Therefore, future work aimed at understanding the mechanisms through which *F. prausnitzii* acquires and disseminates this seven-gene operon is not only key to comprehending microbial ecology but also crucial for potential dietary or probiotic therapeutic interventions targeting the microbiome.

Our microSLAM analyses were well powered, combining five studies that each include both cases and controls and that come from different geographical regions with different diets and lifestyles. Beyond biological variability, variations in DNA extraction methods, and sequencing library preparation protocols can introduce batch effects in microbiome studies, potentially affecting microbial profiling results [63,64]. We therefore included marker depth as a covariate in our microSLAM models to account for biological and technical variability that may confound the τ and β tests. Hence, detecting many significant strain-disease and gene-disease associations across studies is somewhat surprising and suggests that these associations are robust across the study regions. This supports the conclusion that IBD is truly linked to these components of microbial population structure, rather than an unmeasured confounder. However, we cannot rule out confounding due to the limited amount of publicly available data about the study subjects. As opposed to simply including a PC from the GRM to represent the structure of the population, the population structure component of microSLAM uses the GRM to model the cryptic relatedness between samples and estimate what proportion of the relatedness is associated with the phenotype given the included covariates. In future applications, if we were to have more information about host covariates (e.g., diet, medications, or exercise), then the τ test could be used measure the portion of the population structure that is associated with the phenotype after accounting for these other characteristics of the host.

One key novelty of microSLAM compared to SAIGE and other linear mixed models for diploid genomes is that we compute the GRM using gene presence/absence, rather than SNPs. It is worth noting that same gene presence/absence data is used to estimate the GRM and test for trait-gene. This approach has been shown (with SNP data) to reduce power compared to estimating the GRM with an independent set of markers (e.g., SNPs on other chromosomes) [65]. We explored a similar approach by using SNPs in core genes for GRM and random effect estimation in the β test. But we found that for almost all species in our IBD compendium, the SNP data generated a GRM that was very different from

the gene-based GRM, and hence SNPs were not good markers for estimating the population structure in gene presence/absence. Perhaps this approach could work with more investigation into how to pick SNPs for GRM estimation or with a different GRM distance metric.

There are several limitations to our IBD MWAS. First, as discussed above, publicly available metagenomics data rarely includes detailed information about potentially confounding variables, and hence, these important covariates are not accounted for in our models. With so little meta-data it is important to acknowledge that our IBD analyses could be confounded by unmeasured variables (e.g., diet that selects for certain strains and alters IBD risk), though we attempted to mitigate this risk by including maker depth as a covariate. Beyond confounding, more complete meta-data also would be helpful for understanding the capabilities of our method and for functionally interpreting our IBD findings.

Second, we do not find many individual significant genes within this study. This could be partially due to lack of power, especially for 55/71 species with ≤ 60 samples. A much larger dataset would increase our ability to find IBD associations for strains and genes. It would also enable separate modeling of associations for subtypes of IBD, which may have different microbiome signatures [66]. Our simulations suggested that most species did not have sufficient power for separate Crohn's disease and ulcerative colitis models in our IBD compendium. We did investigate if any microSLAM discoveries were mostly driven by one subtype or the other, and we observed very few examples of trait-strain or trait-gene associations that were attributable to only Crohn's samples or only ulcerative colitis samples. One significant gene cluster from *Phascolarctobacterium faecium* (GUT_GENOME040547_00268) was found to be positively associated with IBD (β test) but was only present in Crohn's patients. In addition, there were three species with a significant trait-strain association (τ test) where less than a third of the IBD patients had ulcerative colitis (*CAG-180 sp000432435, Ruminiclostridium E siraeum, and UBA11524 sp000437595*), meaning most of the signal is from individuals with Crohn's disease in those species. While this finding could indicate that most associations are truly shared, it is more likely that we only had sufficient power to detect associations supported by both subtypes and that other subtype-specific associations remain to be discovered in the future with larger individual cohorts. As researchers move towards testing for strain and gene associations in studies with hundreds or thousands of samples, microSLAM's improved specificity and controlled Type 1 error rate, as compared to glm, will be even more important.

Third, microSLAM relies on accurate pangenome profiling, and hence is subject to the same pitfalls as any metagenomic read-based analysis method. For instance, it is possible that some of our discoveries were driven by cross-mapping [67], where metagenomic reads from the wrong species or gene could create a false signal of gene presence (or absence). We therefore recommend validating significant genes from microSLAM's β test with complementary data. In our investigation of the PTS operon, we confirmed the operon structure and variable presence across strains. This, plus the fact that this operon is predominantly found within *F. prausnitzii* and not widely distributed in other species, substantially alleviates concerns about cross-mapping in this analysis. For this validation analysis, we used high-quality genome assemblies, including both isolates and MAGs. MAGs, including those assembled from the sample itself, are particularly useful for strains and species that are underrepresented in genome databases. But MAGs with low completeness or high contamination should be avoided, because they may misrepresent operon structures or fail to capture genes present in the sample but not incorporated into the assembly. Another challenge microSLAM shares with other metagenomic profiling methods is that many of the genes we identified were not annotated, leading to difficulty completing in-depth functional enrichment analyses across species. We did see several consistent trends, such as mobile elements being discovered in multiple species, but lacked power to assess this pattern's statistical significance.

In conclusion, we view microSLAM as an important advance towards identifying microbiome functions that vary within species in association with host traits. The ability of microSLAM to detect associations for species whose relative abundance is not correlated with host traits and to accurately disentangle associations of individual genes versus groups of strain-defining genes make it a useful new hypothesis generating tool for microbiome research. The genes identified by

microSLAM are strong trait biomakers and also interesting candidates for downstream in vitro and in vivo experiments aimed at test causal hypotheses about the roles of specific variable genes, pathways, and strains in host physiology.

## Methods

### Compendium of IBD/healthy case/control metagenomic studies

We compiled a total of 2625 publicly available paired-end shotgun metagenomic samples, sourced from five studies related to either inflammatory bowel disease (IBD) or the Human Microbiome Project (HMP2) and having an average read count greater than 20 million (accession numbers: PRJNA400072, PRJNA398089, PRJEB15371, PRJEB5224 and PRJEB1220). A stringent sample selection process was implemented to ensure (1) all samples included comprehensive metadata, such as disease status, age, and antibiotic usage; and (2) only one sample was selected per subject, considering that multiple time points could have been sequenced from the same subject. Specifically, for multiple time point samples from HMP2, we adopted the same selection criterion used by Lloyd-Price et al [68], selecting week 20 or greater for all subjects, the maximum read count for healthy subjects and the time point with the highest dysbiosis score for IBD patients. The first time point was chosen for the MetaHit project [1,69,70]. Ultimately, a total of 710 samples met these criteria and were included for downstream analysis (S1 and S6 Tables).

### Bioinformatics analysis

Preprocessing of the downloaded metagenomic sequencing libraries was performed using a QC pipeline that includes the following steps: (1) Adapter removal and quality trimming: adapter sequences were removed and low-quality reads were trimmed using Trimmomatic (v 0.39) [71]. (2) Human contamination removal: reads were aligned against the complete human reference genome (CHM13v2.0) [72] and a collection of 2250 genomes known to be contaminated by human sequences [73], using Bowtie2 (v 2.5.1) [74] to identify and remove human contamination. (3) Low-complexity read filtering: low-complexity reads were filtered out using BBduk (v 37.62) [75]. This step involved removing reads with an average entropy less than 0.5, with entropy k-mer length of 5 and a sliding window of 50 (parameters: entropy = 0.5, entropy-window = 0.5, entropyk = 4). Additionally, reads shorter than 50 base pairs (bp) post-filtering were removed (parameters: minlen = 50). (4) Quality reporting: a quality report of the cleaned-up reads was generated with FastQC (v 0.12.1) [76]. After preprocessing, samples with read counts lower than 1 million were removed, resulting in 710 high-quality samples retained for further analysis.

### Pangenome profiling using MIDAS v3

To determine which species within the 710 samples are both prevalent and sufficiently abundant for pangenome profiling, we implemented a two-step analysis using MIDAS v3 [11]. In the first step, we quickly scanned each sample to detect the presence of species by assessing the vertical coverage of 15 universal single-copy marker genes across 3956 distinguishable species in the UHGG v2 database [1,77]. In the second step, we adopted a whole-genome read alignment-based methodology to quantify the abundance of each species. This involved running MIDAS's single-nucleotide variant (SNV) pipeline for species that meet specific criteria from the first step: a median marker coverage of at least 2X and at least 50% of the marker genes uniquely covered. We further classify a species as present based on whole-genome based species abundance estimation, applying the criteria of horizontal coverage > 0.4 and vertical coverage > 5. We further excluded sample-species pairs where the ratio of genome-wide vertical coverage to single-copy marker gene coverage exceeded 4, which helps us to eliminate potential false positives caused by cross-mapping of reads among closely related species and conserved gene families. This stringent criterion also improves computational efficiency [67]. After implementing the aforementioned filtering, 71 species that were present in more than 60 samples and met the abundance criteria were selected for subsequent pangenome profiling analysis. There were 619 samples with at least one species present.

To perform pangenome profiling, we utilized the Genes module from MIDAS v3 [11], which features careful curation of the pangenome database and comprehensive functional annotation. Specifically, a single Bowtie2 index was built for all 71 species, and QC-ed paired-end reads for each sample were aligned to this index. Our analysis included only genes covered by at least 4 reads (--read_depth 4). Genes with an estimated copy number greater than 0.4 were classified as present (--min_copy 0.4). This threshold was selected based on exploratory data analysis and simulations previously performed by our lab. The resulting sample-by-gene presence/absence binary matrix was then used for subsequent association analysis with microSLAM, excluding core genes (absent in less than 10 samples) and rare genes (present in less than 30 samples).

**microSLAM runtime checklist**

1. High-Quality Metagenomic Data

Because proper QC is critical for reliable read-mapping based pangenome profiling, we performed bioinformatics quality control to the raw metagenomic sequencing data, including removing low-quality reads, filtering out low-complexity reads, and removing host reads by aligning to an updated version of the reference genome (e.g., CHM13v2.0). We also removed samples with low post-processed read count (< 1 M reads). Users may follow our QC procedure (details in the Bioinformatics section under the Methods section) or use their preferred QC pipeline.

2. Pangenome Profiling using MIDAS v3.

The following steps outline how to produce the sample-by-gene presence/absence matrix from raw metagenomic shotgun reads using MIDAS v3. However, other genotyping pipelines may also be used as alternatives, such as PanPhlAn 3 [21] and Roary [12]. The required input for microSLAM consists of two tables: (1) a metadata table specifying traits and covariates for each sample, and (2) a sample-by-gene presence/absence matrix that includes accessory genes with sufficient variation across samples, as well as samples with sufficient abundance for the given species.

Step 1: Sample Selection via Species Profiling
Purpose: To identify abundant species suitable for downstream ogene profiling.
Use run_species from MIDAS v3 to estimate species abundance using 15 universal single-copy marker genes. Retain species/sample pairs that meet these criteria: median_marker_depth > 2X and unique_fraction_covered > 0.5.
In this study, we also ran the SNPs workflow of MIDAS v3, which introduced an additional sample selection filter requiring whole-genome vertical coverage ≥ 5X and horizontal coverage ≥ 0.4. While this enhances data quality, it also increases computational demands, making it an optional step for users based on their available resources.

Step 2: Pangenome Profiling
Purpose: To quantify gene coverage.
Use run_genes from MIDAS v3 to profile the abundance of genes on a species level with the following parameters: --read_depth 0.4 --min_copy 0.4. For non-gut metagenomes, use the database `--midasdb_name gtdb`. We included species present in at least 60 samples to ensure statistical power.

Step 3: Prepare Sample-by-Gene Presence/Absence Matrix
Purpose: To convert gene coverage to presence/absence.
Run merge_genes.py from MIDAS v3 with minimal filtering (--genome_depth 0.01, --sample_counts 60). Define genes as present in a sample using the following criteria: minimum read depth of 4 (--read_depth 4); copy number threshold ≥ 0.4 (--min_copy 0.4). Filter genes with too little variation: exclude core genes (absent in less than 10 samples) or rare genes (present in <30 samples). Functional annotations for all the genes can be accessed through MIDASDB v3.

Step 4. Covariates & Trait Data
Purpose: To prepare sample-level covariates and trait data.

To control for confounding, we recommend including appropriate covariates, such as: age, species abundance. For binary traits, be cautious with severely unbalanced classes to minimize bias. For quantitative traits, confirm that normalization (e.g., inverse normal transformation) produces a near-Gaussian distribution.

Step 5. Format input data for microSLAM

Purpose: To prepare data for input to microSLAM.

The first input is a binary sample-by-gene presence/absence matrix ($n \times p$), where $n$ represents samples (rows), and $p$ represents gene families (columns). Each entry is either 0 (absence) or 1 (presence) for a gene family in a given sample. The second input is a phenotype metadata file with $n$ rows. The first column must be a sample identifier (*sample_name).* This is followed by the trait data and covariates. Consistency check: Ensure that sample names match across both input files.

Step 6. Running microSLAM

Purpose: To perform MWAS using the microSLAM R package.

Use the wrapper function run_microSLAM() to execute the full pipeline for a given species: First, compute the genetic relatedness matrix (GRM); Then perform the τ test once per species to evaluate whether population structure (if present) is associated with the traits of interest. This step identifies strain-level associations. Finally, run the β test for each gene to assess whether individual genes are associated with the trait while adjusting for the GRM. For step-by-step analysis and visualization, refer to the stepwise code provided in the microSLAM GitHub repository (https://github.com/pollardlab/microSLAM).

**Statistical model for strain-trait and gene-trait associations while accounting for population structure in metagenomes**

We present microSLAM, a 3-step modeling procedure for detecting within-species genetic variation associated with host biology (Fig 1A). The inputs to microSLAM, for a given species, are a $p$ x $n$ binary matrix of gene family presence/absence values for $n$ host samples and $p$ gene families, a 1 x $n$ vector of trait values for each sample (binary or quantitative), and optionally a $q$ x $n$ matrix $X$ of data for $q$ covariates. The outputs are a measure of population structure (τ) with a permutation p-value and, for each gene family, a coefficient (β) measuring the gene's association with the trait (e.g., log odds ratio for binary traits and logistic regression) with its local false discovery rate (localFDR) adjusted p-value [78]. Results from different species can be interpreted jointly to identify shared trends in trait-associations, such as enriched pathways (see below).

MicroSLAM fits generalized linear mixed effects models that account for the genetic relatedness of strains of a given species across hosts. In Step 1, an $n$ x $n$ sample genetic relatedness matrix (GRM) is computed from the gene presence/absence matrix. To do so, we create an $n$ x $n$ Manhattan distance matrix and then transform this into relatedness using 1-distance. The GRM is used in Step 2 to test if the species' population structure is associated with the trait, which would indicate that hosts with similar trait values tend to have similar strains. For example, for a case/control study, this step aims to detect species where a subset of related strains confers risk. We call Step 2 the τ test, because population structure is modeled using a parameter τ. In Step 3, random effects estimated from the GRM are used to adjust for population structure in a model that is used to test gene families for associations with the trait beyond simply being present in trait-associated strains. We call Step 3 the β test, because a parameter denoted β is used to quantify gene-trait associations.

In Step 2 (τ test), microSLAM fits a generalized linear mixed model. The trait $y_i : i = 1, ..., n$ is modeled as a function of any covariates $X$ (with coefficients α) and random effects $b_i : i = 1, ..., n$ that are estimated from the GRM. The link function f() is the identity function for normalized quantitative traits (linear regression) or the logit function for binary traits (logistic regression):

$$E[f(y)] = \alpha X + b \tag{1}$$

One key component of fitting Model 1 is estimating τ, the variance on the random effects, which depends on the association of the trait to the GRM. This is done iteratively using the average information restricted maximum likelihood (AI-REML) algorithm from the GMMAT [28] method. From this, we obtain a point estimate of τ, a point estimate of the random effects $b_i$, and a statistic, $T = b^2/N$, that measures how associated the species' population structure is with the trait. This T statistic is derived from [48] and computed using the linear setup from [79]. To assess the statistical significance of T, we randomly permute the trait values $B$ times (e.g., $B = 1000$), repeat model fitting, compute a T statistic for each permutation, and use these as an empirical null distribution to estimate a p-value based on how many of the permuted T statistics exceed the observed T statistic. Species with a significant T statistic have population structure that associates with the trait.

In Step 3 (β test), microSLAM fits a second model using the random effects (b) estimated in Step 2 and the presence/absence vector for each gene family, denoted $g$ (with coefficients $\beta$):

$$E[f(y)] = \alpha X + \beta g + b$$

(2)

Model 2 is fit separately for each gene family within each species. $\beta$ measures the gene's association with the trait given the species' population structure and the covariates. To test the null hypothesis that a given gene's $\beta = 0$, we follow the strategy employed in SAIGE [29], which uses a score statistic and does not re-estimate $\alpha$ or $b$. By only fitting Model 1 once per species, this approach is computationally efficient. Microbiome case/control studies are often unbalanced, for example, when a bacterial species is detected in many more controls than cases. To obtain accurate p-values in this scenario, we approximate the score statistics for testing the null hypothesis that $\beta$ is zero using the Saddle Point Approximation (SPA) of the true distribution, as implemented in SAIGE.

To adjust the resulting p-values for multiple testing, we use localFDR [78], which accounts for the high correlation between gene families (i.e., when genes co-occur across strains) that invalidates methods such as Benjamini-Hochberg FDR [80] or Storey's q-value [81]. We transform SPA p-values into Z-values by dividing by two, multiplying times the sign of the estimated $\beta$ coefficient, and converting the resulting numbers to quantiles. Then, localFDR uses maximum likelihood estimation to approximate the null Z-value distribution and identify Z statistics that deviate from this distribution. We implement this using the *locfdr* v1.1-8 package in R, fitting the null distribution to the Z-values between the 10th and 90th percentiles across all species (S5 Fig).

### Generalized linear model

A standard generalized linear model (glm) [82] was fit for all genes for all species that were analyzed with microSLAM. This is a logistic regression model using *glm* in R with case/control status as the outcome, gene presence/absence as a predictor, and age as a covariate.

### τ test simulations

We performed simulations to assess the false positive rate and power of microSLAM's τ test. To assess the false positive rate (simulation 1), we set up a simulation where a binary trait was generated independently of GRMs. For each of 1000 iterations and n = 100 samples, the trait $y$ was simulated using a binomial distribution with a success probability of 0.5, and a covariate was simulated with a normal distribution centered at 45 with a standard deviation of 15, similar to the age distribution in our IBD compendium. Next, for each iteration, a gene presence/absence matrix was simulated with p = 1000 genes. This included 400 "core" genes simulated from a binomial distribution with a success probability of 0.8, 400 "accessory" genes simulated from a binomial distribution with a success probability of 0.2, and 200 genes simulated based on presence of a strain unrealted to the trait $y$, as follows. The strain's presence/absene across samples was simulated using a binomial distribution with a success probability of 0.5, and then presence/absence for each of the 200 genes was set

to absent if the strain was absent and simulated from a binomial distribution if the strain was present, where the success probabilities were chosen such that the average odds ratio of a given gene being present if the strain is present is 1.8. After the genes are simulated, the GRM is calculated and the population structure test is run with 100 permutations. The p-value is calculated for each iteration as the number of permutations with a more extreme T statistic than the observed T statistic. The false positive rate is calculated as the number of iterations with a p-value <0.05.

The power test (simulation 2) is carried out in a similar fashion, with two key changes. First, we explored a range of sample sizes (n = 60, 100, 250) to assess the relationship between sample size and power. Second, we simulated the trait y based on presence/absence of the simulated strain. Specifically, we explored a range of effect sizes, quantified with an odds ratio $\theta/(1-\theta)$ ranging from 1.0 to 2.5. For a given odds ratio, we set $strain_\delta = (1-strain)*(1-\theta) + stain*\theta$ and then generated the trait $y_{strain}$ using a binomial distribution with success probability equal to $strain_\delta$ : $y_{strain} = rbinom(n, 1, strain_\delta)$. This creates a stronger relationship between the trait and presence of the strain as the odds ratio increases. For each odds ratio and sample size, 125 iterations were run and power was calculated as the proportion of iterations with a significant $\tau$ test divided by 125.

## β test simulations

We performed simulations to assess the false positive rate and power of microSLAM's β test versus a standard glm. To assess false positives (simulation 1), we generated data in which no genes were associated with the trait, so that all genes are false positives. We computed a p-value for each gene using each modeling approach and tracked the proportion of genes with p < 0.05. In order to simulate real population structure, while introducing some random variation, we used the observed GRM for each of the 71 species in our metagenomic compendium to help generate simulated gene presence/absence matrices. Specifically, we first decomposed the observed GRM for each species into its first 10 principal components (PCs). We then standardized each PC by dividing each value by the PC's standard deviation $PC_{std} = PC/sd(PC)$ and computed the standard normal probability for each sample's loading on each standardized PC (one per sample i per PC dimension j): $p_{i,j} = pnorm(PC_{std})$. The probabilities $p_{i,j}$ retain relationships between samples across the 10 dimensions. For each of the 10 PCs, we simulated the presence/absence of 90 genes using a binomial distribution with a success probability equal to the sample's $p_{i,j}$ for that PC, for a total of 900 genes correlated with one dimension of the population structure. We also simulated 100 uncorrelated genes using a binomial distribution with a success probability chosen from a uniform distribution between 0.2 and 0.8. From the resulting 1000 x n gene presence/absence matrix, we simulated a binary trait (y) using the first two PCs (PC1 and PC2), as follows. We set y equal to one in a given sample if its loadings on PC1 and PC2 had opposite signs (either PC1 > 0 and PC2 < 0 or PC1 < 0 and PC2 > 0). This created a nonlinear relationship between y and 180 of the simulated genes (S6 Fig).

For each simulated y and gene presence/absence matrix, we ran microSLAM to compute a GRM and estimate population structure (τ). These new τ values were different from the species' observed τ values and greatly varied across species, as in the observed data (S7 Fig). After the GRM was calculated, and the τ test was run, both glm and microSLAM's β test were run to test for gene-trait associations. The false positive rate was determined by summing the number of genes with p-values < 0.05 divided by the total number of genes, excluding genes simulated from PC1 or PC2 (p = 820 genes).

To assess power for the β test (simulation 2), we start with the data from simulation 1 and set $y_\delta = (1-y)*(1-\theta) + y*\theta$ for an odds ratio of $\theta/(1-\theta)$. Then, we generated presence/absence for 100 additional genes from a binomial distribution with success probability $y_\delta$ : $G\_y = rbinom(n, 1, y_\delta)$. These genes, which we denote as $G_y$, are positives (associated with y) and all other genes are negatives (independent of y). At $\theta = 0.5$ (i.e., an odds ratio of one), the generated genes $G_y$ will not be associated with the trait. At $\theta = 0.55$, the average odds ratio will be 1.2. We investigated $\theta$ values between 0.52 and 0.78. We checked, and the odds ratios across simulations with the same $\theta$ value did not deviate more than 0.1 from the expected values.

We ran glm and microSLAM on each simulated dataset. As expected, the population structure test yielded estimated $\tau$ values that increase notably with the simulated odds ratio (i.e., as the association between y and the genes $G_y$ increases). In order to assess power, we used the negative genes to establish a significance threshold for each modeling approach such that the empirical false positive rate would be no more than 0.05. Applying these thresholds to the positive genes $G_y$, we computed power as the proportion of positive genes detected. Power was compared between glm and microSLAM across odds ratios and species (each with different sample sizes and GRMs).

### Relative abundance test

We calculated the relative abundance of each species by downloading the UHGG v2 Kraken database from MGnify [22] and running Kraken2 [43,45] with options *--paired --minimum-hit-groups 3* and then Bracken [44,45] with options *-l S -t 1000.* We computed relative abundance as a given species' bracken coverage divided by the total coverage, and we removed species with less than 0.05% relative abundance. We then performed logistic regression, using the case/control label as the dependent variable (y) and relative abundance as the independent variable, with age as a covariate. The estimated log odds ratios and p-values from these logistic regression analyses were compared with outputs from the microSLAM $\tau$ test, after using localFDR to adjust p-values at a 0.1 level.

### Identification of seven-gene GFR operon in *F. prausnitzii*

The gene *grfD* from *F. prausnitzii* (UHGG species id 102272) reported as significant by microSLAM's $\beta$ test corresponds to the EIID component and is part of a putative GFR operon that encodes the Fructoselysine PTS system. A similar gene in *S. Typhimurium 14028s* has been identified as responsible for the utilization of fructoselysine [58,83]. To determine whether this EIID gene is part of a gene cluster that forms an operon - that is, genes that are sequentially arranged on the chromosome and co-regulated - we conducted the following analysis. First, we retrieved the neighboring genes upstream and downstream of this EIID gene in the UHGG reference genome for this *F. prausnitzii* species, considering up to five genes in each direction. We then used blastn [84] to identify homologous regions in three different sets of genomes: (1) 85 NCBI *F. prausnitzii* genomes, (2) all genes in the nine UHGG *F. prausnitzii* species clusters including metagenome-assembled-genomes (MAGs), and (3) all MAGs in the 71 species that were investigated in this study. These five genes from *F. prausnitzii* were also aligned to the corresponding *S. Typhimurium* operon (gfrABCDF) using Blastp [84]. All genomes were annotated using Prokka [85]. Genes were annotated with BlastKOALA [55] and eggNOG-mapper [86].

### Selection of *Faecalibacterium prausnitzii* genomes

To avoid incorrectly assessing the seven-gene GFR operon as incomplete in an assembly simply due to fragmented contigs, we only selected high-quality *F. prausnitzii* genomes with assembly levels of scaffold, chromosome, or complete genome, and we specifically excluded atypical genomes. In total, we downloaded 105 genomes of *F. prausnitzii* from NCBI (using the taxon identifier 853). We assessed the genome quality using CheckM [87], and retained only genomes that met the following criteria: completeness >= 90, contamination <=5, and strain heterogeneity <= 10. After this filtering, 85 *F. prausnitzii* NCBI genomes were retained for the GFR operon screening analysis (S4 Fig). Next, we used dRep [88] to perform pairwise genome comparisons based on Average Nucleotide Identity (ANI). This dRep analysis involved first clustering all the genomes using the Mash heuristic for ANI [89] and subsequently using MUMMER [90] to compute ANI on sets of genomes that have at least 90% Mash ANI before performing a secondary clustering. As a result, 52 secondary clusters were formed at 98% MUMMER ANI (-comp 90 -con 5 -pa 0.95 -sa 0.98 -nc 0.8). Hierarchical clustering of the 52 representative genomes using average linkage was performed using the pairwise MASH similarity matrix ('scipy.cluster.hierarchy' package).

In addition to NCBI Genomes, we also collected all nine *F. prausnitzii* species clusters from UHGG v2 [1], using the same selection criteria to ensure assembly quality. We calculated the pairwise genome similarity between the resulting 52 *F. prausnitzii* genomes and the nine representative genomes from UHGG *F. prausnitzii* species using fastANI [91]. We also compared each NCBI *F. prausnitzii* genome to the nine representative genomes from UHGG, and each NCBI genome was assigned to the UHGG species cluster with the highest ANI (ANI >= 95%). If this similarity level was not reached, the NCBI genome remained unassigned. Eight *F. prausnitzii* species in UHGG were represented by the 52 NCBI *F. prausnitzii* genomes. The information for the 85 genomes is available in S5 Table.

## Supporting information

**S1 Text.  Additional methodological details.**
(DOCX)

**S1 Table.  IBD compendium data information.**
(TSV)

**S2 Table.  IBD compendium species results.**
(TSV)

**S3 Table.  IBD compendium gene family results.**
(TSV)

**S4 Table.  PTS operon Blast KOALA results.**
(TSV)

**S5 Table.  NCBI *F. prausnitzii* genomes.**
(TSV)

**S6 Table.  Quality control for all samples.**
(XLSX)

**S1 Fig.  GLM and microSLAM β test power evaluations for 71 simulated species.** These plots show estimated power of β tests using data from simulation 2 in which a gene presence/absence matrix and binary trait were simulated based on the observed GRMs from the 71 species in the IBD compendium using a range of different effect sizes (odds ratios, horizontal axes). There is one panel per GRM (labeled with species ID), and panels are ordered from lowest to highest sample size. Power was computed as the proportion of positive genes discovered at an empirical localFDR of 0.05 for both microSLAM (*red*) and glm (*blue*). As the number of samples increases there tends to be a larger difference between the glm and the microSLAM models.
(TIFF)

**S2 Fig.  P-values from microSLAM's β test are somewhat conservative, while GLM's are inflated.** Q-Qplots of β Test results for a simulation with positive genes ($G_y$; *pink circles*: microSLAM, *light blue circles*: glm) plus negative genes that are linked to a strain (strain; squares) or randomly generated (other; triangles) (β test simulation 3, **S1 Text**). A) Compared to glm (blue), microSLAM (red) better distinguishes the positive genes $G_y$ from those simulated from the strain. The number of positive genes was one (*left*), two (*middle*), or three (*right*). The value of τ increases with each additional gene $G_y$. B) As the relationship between the strain and *y* is increased (left to right), the value of τ increases, and the rate of inflation increases for glm. Across different values of τ, microSLAM remains slightly conservative and continues to rank the positive genes $G_y$ highest, indicating high specificity.
(TIFF)

**S3 Fig. Exploration of batch effects.** (**A**) Median coverage of 15 universal, single-copy genes ("marker depth") in each sample in the IBD compendium, stratified by cohort. We computed marker depth using the MIDAS v3 genes flow. Four representative species are shown. (**B**) Heatmap of random effect values for each of 71 species across all samples where it was detected in the IBD compendium. The red-to-blue color scale denotes the association between strains and IBD (binary case/control status). Red: strains positively associated with IBD; Blue: strains negatively associated with IBD. The study and IBD subtype of each sample are shown on the left. IBD subtypes: Blue = control, red = Crohn's disease (CD), yellow = Ulcerative colitis (UC). CD and UC were combined as cases in the microSLAM modeling. Studies: Franzosa (NCBI BioProject PRJNA400072; orange), He (PRJNA398089; pink), Nielsen (PRJEB15371; blue), HMP2 (PRJEB5224; green), MetaHIT (PRJEB1220; yellow). Species are ordered by the standard deviation of *b* (left = highest standard deviation), where higher standard deviation indicates greater strain diversity that is associated with case/control status. The samples in each column are ordered by lowest to highest average b value. A few studies (e.g., Nielsen) have more controls than others, but there is no systematic relationship between study and population structure. CD and UC tend to have similar distributions of *b* values (i.e., red and yellow are mixed on the left side bar). While we cannot rule out confounders that were unmeasured in the publicly available data that we could access, these patterns suggest that our findings are not obviously biased by differences in study population (e.g., diet, medical care, geography, type of IBD) that could confound measured associations between case/control status and microbiome strains and genes.
(TIFF)

**S4 Fig. *F. prausnitzii* PTS operon evolves as a unit across diverse genomes.** This operon comprises seven genes (occasionally eight genes) that were consistently present or absent together across 53% (49/85) of *F. prausnitzii* genomes from NCBI. The order and orientation of genes in the operon is conserved. This heatmap shows the genes (rows; position 0 is *gfrD,* which was significant after localFDR adjustment of microSLAM β test p-values). The other genes were significant before localFDR adjustment and are indexed relative to *gfrD* in the heatmap. Columns represent 423 contigs from 85 *F. prausnitzii* high-quality NCBI genomes. The color of the heatmap shows the blastn sequence similarity of the gene sequence in the contig compared to the sequence in the *F. prausnitzii* reference genome used in our microSLAM analysis (red=highest similarity, white=no significant match). The seven genes in the operon (middle rows of the heatmap) have high sequence similarity when they are present and are present together (red on left), whereas flanking genes are more variably present and have lower sequence identity (blue in top and bottom rows).
(TIFF)

**S5 Fig. LocalFDR p-values and Z-values.** (**A**) Histogram of p-values for microSLAM's β test (left) and glm (right). (**B**) Output from localFDR showing the distribution of the null z-values (green) versus the distribution of the z-values that do not follow the null (pink). Yellow triangles denote the z-value thresholds corresponding to a localFDR of 0.2.
(TIFF)

**S6 Fig. Example of data and results from microSLAM β test Simulation 1.** (**A**) Simulated gene presence/absence matrix based on the GRM of *Bacteroides thetaiotaomicron* plotted as a heatmap (grey = gene present in a given sample, white = gene absent). Genes are in columns and are labeled according to how they were simulated (0 = random, pc1–10 = using one of the first 10 principal components of the observed GRM for *B. thetaiotaomicron*. This presence/absence matrix has some population structure (estimated τ = 2.30), but no genes were simulated to be associated directly associated with the trait which is defined by the first two PCs. (**B**) Q-Qplot of p-values from all genes not from PC1 or 2 from microSLAM's β test (red) and glm (blue) applied to the simulated gene presence/absence matrix in (A). There is a much higher error rate for the glm model. On the other hand, microSLAM is overly conservative (i.e., underpowered). (**C**) Q-Q plot for microSLAM's β test (red) and glm (blue) applied to the observed *B. thetaiotaomicron* gene presence/absence matrix from the IBD compendium. The trends are very similar to those in the simulation.
(TIFF)

**S7 Fig. β Test simulation τ values versus observed τ values in IBD compendium.** In the β Test Simulation 1 and 2 set up, we generated gene presence/absence matrices using the observed GRMs for the 71 species in the IBD compendium. Our objective was to generate simulated data that was similar to but not identical to the observed data (**Methods**). This scatter plot shows the τ values estimated by microSLAM on the simulated data (*y axis)* compared to the corresponding τ values estimated from the real data in the IBD compendium (*x axis*). The n τ from the simulation cover a similar range of values as those from the real data while not being highly correlated.
(TIFF)

## Acknowledgments

We would like to thank Byron Smith, Abigail Lind, Xiaofan Jin, Lei Liu, and Eran Segal for their insightful discussions about the project. The microSLAM R package is available at https://github.com/pollardlab/microSLAM.

## Author contributions

**Conceptualization:** Miriam Goldman, Katherine S. Pollard.

**Data curation:** Miriam Goldman, Chunyu Zhao.

**Formal analysis:** Miriam Goldman, Chunyu Zhao.

**Funding acquisition:** Katherine S. Pollard.

**Investigation:** Miriam Goldman, Chunyu Zhao, Katherine S. Pollard.

**Methodology:** Miriam Goldman, Chunyu Zhao, Katherine S. Pollard.

**Project administration:** Katherine S. Pollard.

**Resources:** Katherine S. Pollard.

**Software:** Miriam Goldman.

**Supervision:** Katherine S. Pollard.

**Validation:** Miriam Goldman, Chunyu Zhao.

**Visualization:** Miriam Goldman, Chunyu Zhao, Katherine S. Pollard.

**Writing – original draft:** Miriam Goldman, Chunyu Zhao, Katherine S. Pollard.

**Writing – review & editing:** Miriam Goldman, Chunyu Zhao, Katherine S. Pollard.

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
