## [Decision Letter · Decision Letter 0]

13 Dec 2024

PCOMPBIOL-D-24-01068

Improved detection of microbiome-disease associations via population structure-aware generalized linear mixed effects models (microSLAM)

PLOS Computational Biology

Dear Dr. Pollard,

Thank you for submitting your manuscript to PLOS Computational Biology. After careful consideration, we feel that it has merit but does not fully meet PLOS Computational Biology's publication criteria as it currently stands. Therefore, we invite you to submit a revised version of the manuscript that addresses the points raised during the review process.

Please submit your revised manuscript within 60 days Feb 12 2025 11:59PM. If you will need more time than this to complete your revisions, please reply to this message or contact the journal office at ploscompbiol@plos.org. Please include the following items when submitting your revised manuscript:

* A rebuttal letter that responds to each point raised by the editor and reviewer(s). You should upload this letter as a separate file labeled 'Response to Reviewers'. This file does not need to include responses to formatting updates and technical items listed in the 'Journal Requirements' section below. Please pay special attention to the comments of reviewer #1 regarding the statistical framework and rigor of your work.

We look forward to receiving your revised manuscript.

Kind regards,

Iddo Friedberg, Ph.D.

Academic Editor

PLOS Computational Biology

Rob De Boer

Section Editor

PLOS Computational Biology

Feilim Mac Gabhann

Editor-in-Chief

PLOS Computational Biology

Jason Papin

Editor-in-Chief

PLOS Computational Biology

**Journal Requirements:**

3) We notice that your supplementary Figures are included in the manuscript file. Please remove them and upload them with the file type 'Supporting Information'. Please ensure that each Supporting Information file has a legend listed in the manuscript after the references list.

Potential Copyright Issues:

- Figure 1; Please confirm whether you drew the images / clip-art within the figure panels by hand. If you did not draw the images, please provide a link to the source of the images or icons and their license / terms of use; or written permission from the copyright holder to publish the images or icons under our CC BY 4.0 license. Alternatively, you may replace the images with open source alternatives. See these open source resources you may use to replace images / clip-art:

5) Please ensure that the funders and grant numbers match between the Financial Disclosure field and the Funding Information tab in your submission form. Note that the funders must be provided in the same order in both places as well.

**Reviewers' comments:**

Reviewer's Responses to Questions

**Comments to the Authors:**

Reviewer #1: First of all, I like the main idea of the manuscript, which applies statistical methods from GWAS studies to metagenomics—a novel approach that I believe will be useful for researchers in microbiome studies.

That said, the manuscript currently falls short in clarity, particularly concerning the statistical model, which is the primary focus of the work. The model is challenging to understand due to confusing symbols and unexplained technical terms across the manuscript. The methodology part of the manuscript is hard to read. Below are some examples:

For example, the author first mentioned tau that refers to population structure in line 571, 584 (tau test actually mentioned at the beginning of the result, line 127). However, it is only in line 594 that the author clarifies that tau represents the variance of the random effects.

Line 611 is another example: “Similar to the strategy used in SAIGE [27] , we directly calculate the score statistic for each gene by fitting the covariate and population structure adjusted genotype vector to the phenotype. Doing a direct computation given the random effect is an efficient strategy to reduce compute time;” Terms like “score statistic” and “direct computation” are presented here but without explanation. Likewise, it’s unclear what is meant by “fitting the covariate- and population-structure-adjusted genotype vector to the phenotype.” These terms seem directly drawn from the SAIGE paper, but readers won’t understand them without reading the SAIGE paper.

Compared to the confusing main text, the supplementary text is much better but still not clear. In general, I had to rely heavily on the references cited within the manuscript to fully understand the methodology.

The main statistical model is clearly closed to SAIGE [27] and GMMAT [26], which is okay even if the same method is used. However, the authors do not clearly indicate what they have adapted from previous work and what is new in their own implementation of GLMM. If authors did modify the original method, a comparison between the new framework and SAIGE or GMMAT would be needed.

For further improvement, I would recommend collaborating with a statistician or biostatistician to rewrite the methodology section. Alternatively, the authors might consider revising the manuscript as an analysis paper that highlights the novel analysis pipeline.

Reviewer #2: It was a pleasure to read the manuscript from Goldman and Co,. The language is clear and well written. I have to admit, that I have no experience yet with running pangenome analysis on populations of different species found in metagenomes with tools such as MIDAS2 or 3. So this was a bit of a learning curve for me, to read this manuscript.

I believe the whole approach of identifying microbiome species and their genes that have a significance in the occurrence of disease is really interesting. The microSLAM package seems to do that pretty well and I was pleasantly surprised by the finding of genes (the fructoselysine operon) that seem to be more mobile than one expects from a group of genes involved in carbon metabolism.

The structure of your manuscript is good in my eyes and I like the use of publically available datasets instead of using artificial datasets to show how the method works. That makes this manuscript on an R-package a lot more interesting for a microbiologist as myself.

For this manuscript, I find that I have very little to comment on and below you only find a comment on the installation instructions of microSLAM and several minor comments.

Installation comment.

One of the first things I did was to try to install microSLAM in R-studio. It turned out, that I needed an extra library in order to be able to install microSLAM with the command install_github which was not on your github instructions. That library is "remotes". check if that is needed. After that, the installation went fine, and I could run the example that you made in the MicroSLAM repository.

Minor comments.

Lines: 276 - 278: This sentence seems rather vague and is not complete in my opinion. There are DNA extraction methods which can cause batch effects, as well as sequencing library preparation methods. Do you have a reference for that statement?

Line: 329-330 This reads weird: " ... addition to the understand these genes ..."

Line 469 - 470: Here is something missing or not right in the text. The sentences do not make sense to me.

Line 484 - 487: I wonder, could you also use MAGs derived from the same samples to test these findings? For instance identifying the gene operon from the MAGs? Or is could that be fraught with other issues?

Reviewer #3: The current study describes microSLAM a novel method for carrying out microbiome-phenotype association studies. While the work first pinpoints individual trait-associated strains, its novelty is the focus on the specific microbial genes associated with the trait. This is an important improvement over the state of the art, as microbiome HGT requires a fairly loose definition of a strain and makes identification of culprits difficult (and often non-reproducible) across studies. The paper is well written and timely. However, some problems of the manuscript are detailed below:

Major

1. The stated goal of the study seems to be the finding of trait causality rather than biomarkers. If indeed so, it is unclear why the associated organism needs to be pinpointed first (tau test). This point of view suggests that “genes” and not “functions” are the basic units of causality. However, individual genes, specific to bacterial lineages, may or may not reflect all of the specific function F within the microbiome, which these genes carry out. That is, species A and unrelated species B may perform the same function F using different genes. In the absence of the function to phenotype mapping, identified gene-phenotype associations are incomplete and biased by our ability to reconstruct strains. I would suggest that some annotation of metagenomic genes (or directly reads!) with associated functions (e.g. EC numbers using HUMANN or MGNIFY pipeline, or a wider set of fusion functions using fusionDB or mi-faser) would be more relevant for the type of analysis that microSLAM wants to bring forth. Incidentally, not requiring species baselines would allow for a much wider analysis of genes/functions in the microbiome.

I realize that the above analysis may not be possible given the ready-made tool. However, I insist on the inclusion of some discussion of the merits in establishing causality of the described approach vs. a function-focused, population structure one.

If finding biomarkers (not causality) is the goal – this should be made more clear in the text.

2. It is unclear how the identified associations can be verified. While simulations are meaningful, they do not capture the specific biology of disease. In the absence of experimental data (and limited gene info as in Fig 4E), an option for validation is to separate the development set of metagenomes into individual cohorts (and do cross-validation) or to collect additional data to explore whether the strains and/or genes identified for one set can be extracted for the others. How reproducible are the results is an important question, unless the claim is that IBD in different cohorts of people is caused by different genes. If this is indeed the claim, it should be made significantly more clear.

Minor

1. I am curious how microSLAM performs across vs within different cohorts, where batch effects may be an issue. In the analysis, 710 metagenomes form 5 studies are used. Would using one be better? Was there any error correction? Some discussion on the topic of sequencing depth, techniques differences, etc. in establishing microbiome structure would be helpful. On that same note, is there ever a time when two different cohorts studying the same disease can be unified for more power?

2. There are five issues/observations/concerns with using microSLAM that were raised by the authors in the paper discussion. These make using the tool difficult in a generic situation, i.e. I have a bunch of metagenomes and I want to know which genes are associated with a metagenome-specific trait. Please provide a specific description (almost like a help document) for what is expected as the data input so that the results can be trusted.

3. I would argue that the word several here “several human diseases [2–5]” is not necessary. It implies that only a few diseases are microbiome associated, but I would argue that many are.

**Have the authors made all data and (if applicable) computational code underlying the findings in their manuscript fully available?**

Reviewer #1: Yes

Reviewer #2: Yes

Reviewer #3: Yes

PLOS authors have the option to publish the peer review history of their article (what does this mean? ). If published, this will include your full peer review and any attached files.

**Do you want your identity to be public for this peer review?** For information about this choice, including consent withdrawal, please see our Privacy Policy .

Reviewer #1: No

Reviewer #2: **Yes: ** Thomas H.A. Haverkamp

Reviewer #3: No

**Figure resubmission:**
---

## [Decision Letter · Decision Letter 1]

13 May 2025

Dear Dr. Pollard,

We are pleased to inform you that your manuscript 'Improved detection of microbiome-disease associations via population structure-aware generalized linear mixed effects models (microSLAM)' has been provisionally accepted for publication in PLOS Computational Biology.

Best regards,

Iddo Friedberg, Ph.D.

Academic Editor

PLOS Computational Biology

Rob De Boer

Section Editor

PLOS Computational Biology

Reviewer's Responses to Questions

**Comments to the Authors:**

Reviewer #1: I am satisfied with the revision and the manuscript has been largely improved. There are some printing errors for tau and beta across the manuscript.

Reviewer #3: Thank you for making the changes. The manuscript looks more clear to me now

**Have the authors made all data and (if applicable) computational code underlying the findings in their manuscript fully available?**

Reviewer #1: Yes

Reviewer #3: None

PLOS authors have the option to publish the peer review history of their article (what does this mean? ). If published, this will include your full peer review and any attached files.

**Do you want your identity to be public for this peer review?** For information about this choice, including consent withdrawal, please see our Privacy Policy .

Reviewer #1: No

Reviewer #3: No

---

## [Editor Report · Acceptance letter]

PCOMPBIOL-D-24-01068R1

Improved detection of microbiome-disease associations via population structure-aware generalized linear mixed effects models (microSLAM)

Dear Dr Pollard,

I am pleased to inform you that your manuscript has been formally accepted for publication in PLOS Computational Biology. Your manuscript is now with our production department and you will be notified of the publication date in due course.

With kind regards,

Zsofia Freund
